# Predicting Angiogenesis by Endothelial Progenitor Cells Relying on In-Vitro Function Assays and VEGFR-2 Expression Levels

**DOI:** 10.3390/biom9110717

**Published:** 2019-11-08

**Authors:** Nadin Sabbah, Tal Tamari, Rina Elimelech, Ofri Doppelt, Utai Rudich, Hadar Zigdon-Giladi

**Affiliations:** 1Laboratory for Bone Repair, Rambam Health Care Campus, Haifa 3109600, Israel; nadin.tech@gmail.com (N.S.); frumzi@gmail.com (T.T.); dr.rinaperio@gmail.com (R.E.); ofridoppelt@gmail.com (O.D.); 2The Ruth and Bruce Rappaport Faculty of Medicine, Technion-Israel Institute of Technology, Haifa 3109601, Israel; utairudich@gmail.com; 3Department of Periodontology, Rambam Health Care Campus, Haifa 3109601, Israel

**Keywords:** human endothelial progenitor cells, angiogenesis, potency assay, conditioned medium, chemoattractant, mesenchymal stem cells, vascular endothelial growth factor receptor 2 (VEGFR-2)

## Abstract

Clinical trials have demonstrated the safety and efficacy of autologous endothelial progenitor cell (EPC) therapy in various diseases. Since EPCs’ functions are influenced by genetic, systemic and environmental factors, the therapeutic potential of each individual EPCs is unknown and may affect treatment outcome. Therefore, our aim was to compare EPCs function among healthy donors in order to predict blood vessel formation (angiogenesis) before autologous EPC transplantation. Human EPCs were isolated from the blood of ten volunteers. EPCs proliferation rate, chemoattractant ability, and CXCR4 mRNA levels were different among donors (*p* < 0.0001, *p* < 0.01, *p* < 0.001, respectively). A positive correlation was found between SDF-1, CXCR4, and EPCs proliferation (R = 0.736, *p* < 0.05 and R = 0.8, *p* < 0.01, respectively). In-vivo, blood vessels were counted ten days after EPCs transplantation in a subcutaneous mouse model. Mean vessel density was different among donors (*p* = 0.0001); nevertheless, donors with the lowest vessel densities were higher compared to control (*p* < 0.05). Finally, using a linear regression model, a mathematical equation was generated to predict blood vessel density relying on: (i) EPCs chemoattractivity, and (ii) VEGFR-2 mRNA levels. Results reveal differences in EPCs functions among healthy individuals, emphasizing the need for a potency assay to pave the way for standardized research and clinical use of human EPCs.

## 1. Introduction

In the last decade, there has been growing interest in clinical research using autologous transplantation of endothelial progenitor cells (EPCs) as a new strategy to achieve therapeutic angiogenesis in humans. Safety and efficacy of autologous EPC therapy has been demonstrated in clinical studies of infarcted myocardium in humans [1,2,3,4,5]. The clinical use of autologous EPCs transplantation was described in many medical fields including: acute cerebral infarct [6] diabetic foot [7], critical limb ischemia [8], liver cirrhosis [9] and traumatic bone defects [10]. Angiogenesis is important to maintain the integrity of tissue perfusion, which is crucial for physiologic organ function. It is well known that EPCs’ function and number is impaired due to certain circumstances, such as diabetes [11,12]. The reduction and dysfunction of circulating EPCs have been reported in both type I and type II diabetic patients. On the other hand, an increased number of circulating EPCs was found amongst cancer patients [13,14,15], smokers [16,17], and obstructive sleep apnea patients [18].

The clinical efficacy of autologous cell therapies may vary among patients, in part, due to different genetic and physiologic status and medical conditions. Similarly, clinical trials using autologous mesenchymal stem cells (MSCs) for the treatment of different medical indications show conflicting results. For example, patients with Crohn’s disease treated with autologous MSCs showed heterogeneous clinical outcomes. In vitro—expanded MSCs presented similar morphology, phenotype, and growth potential; however, following MSCs infusion, two patients showed improvement and three presented disease worsening [19]. Since previous studies clinically examined autologous EPCs transplantation, and demonstrated changes in cell number and function in various pathological conditions, it is essential to find genes and/or in-vitro assays that can predict the angiogenic potential of each individual before cell transplantation. We hypothesize that EPCs isolated from unrelated healthy donors will show different biological functions due to genetic differences that may influence therapeutic outcome. Therefore, the aim of this study was to find genes and/or in-vitro assays that can predict blood vessel formation by EPCs.

## 2. Materials and Methods

There were two major parts to this study: (1) in-vitro studies that investigated the phenotypic and genetic characteristics of EPCs from healthy donors, and (2) in-vivo studies that investigated angiogenic potential of these cells in a subcutaneous mouse model. All experiments were performed in accordance with Rambam Health Care Campus’s Helsinki committee for human experiments (Helsinki number 0397-12 RMB) and the Committee for the Supervision of Animal Experiments at the Ruth and Bruce Rappaport Faculty of Medicine, Israel Institute of Technology-Technion (approval #IL118-1-2015). All methods were performed in accordance with the relevant guidelines and regulations.

### 2.1. Isolation and Culturing of Endothelial Progenitor Cells from Human Peripheral Blood

EPCs were isolated and cultured as previously described [20]. Briefly, human EPCs were isolated from the blood of 18 healthy donors who signed informed consent (Helsinki number 0397-12 RMB). For cell isolation, we collected 50 mL of peripheral blood into sterile heparinized tubes. Blood was diluted 1:1 with phosphate-buffered saline (PBS). Mononuclear cells (MNCs) were isolated using density gradient centrifugation (Lymphoprep, Axis-Shield Diagnostics Ltd., Dundee, Scotland) and pelleted cells resuspended in EGM-2 SingleQuote (Lonza Walkersville Inc. Walkersville, MD, USA), an endothelial growth medium containing 20% fetal bovine serum (FBS) and penicillin-streptomycin (Biological Industries Ltd., Beit Haemek, Israel). Cells were seeded on six-well plates coated with 5 µg/cm^2^ of fibronectin (Biological Industries Ltd.) and grown at 37 °C with humidified 95% air/5% CO_2_. After 4 days of culture, nonadherent cells were discarded by gentle washing with PBS, and fresh medium was applied. The attached cells were continuously cultured with EGM-2. Cells were fed three times per week then split when they reached ~80% confluence by brief trypsinization, using 0.5% trypsin in 0.2% ethylenediaminetetraacetic acid (EDTA).

### 2.2. Characterization of Human EPCs

In order to identify human EPCs, 2 × 10^5^ cells were suspended in 50 μL fluorescence-activated cell sorting (FACS) buffer (PBS containing 0.5% FBS) then stained with 0.2 mg/mL of each of the following anti-human antibodies: CD31-PE (LifeSpanBioSciences, Seattle, WA, USA), CD34-BV421 (BD Biosciences, San Jose, CA, USA), CD45-BB515 (BD Biosciences), VEGFR-2 PE (R&D Systems, Minneapolis, MN, USA), CD14-FITC (BD Biosciences), and CXCR4-PE (BioLegend, San Diego, CA, USA). OneComp eBeads (Carlsbad, CA, USA) were first stained with 1 μL of each different fluorochrome then used as single-color compensation controls. An unstained sample served as a reference for positive staining. Cells were analyzed using a Cyan flow cytometry (Beckman Coulter, Brea, CA, USA). Single cell data was analyzed using FlowJo, LLC software that gave a percentage for positive cells.

### 2.3. Cell Proliferation Assay XTT

We used Cell Proliferation XTT Kit (Biological Industries Ltd.). Briefly, 1 × 10^4^ human EPCs in 100 μL EGM-2 were seeded onto 96 well plates in triplicate. The absorbance was examined every 24 h and measured with an ELISA plate reader (PowerWave XS2, BioTek, Winooski, VT, USA) at a wavelength of 660 nm subtracted from 475 nm measurements. For each time point, mean value was calculated and expressed as a fold increase according to the slope [21]. Optical density was converted to the number of cells using a standardized calibration curve that is described in the Appendix A.

### 2.4. EPC Conditioned Medium (EPC-CM) Preparation

One million human EPCs were cultured in EGM-2 until 80% confluence. After incubation for 48 h, 10 mL medium were collected and concentrated using a centrifugal filter (Merck Millipore, Tullagreen, Ireland).

### 2.5. Migration Assay

The migration assay was performed with an 8 μm pore size millipore chamber (Millicell^®^, Darmstadt, Germany); 2.5 × 10^3^ MSCs were seeded on top of the porous membrane, in 200 μL starvation media DMEM (0.5% FBS). The lower chamber was loaded with human EPCs concentrated condition medium. After overnight incubation, the cells were stained with crystal violet solution (Sigma-Aldrich, Burlington, MA, USA) and the number of cells that migrated to the lower side of the membrane was quantified. The assay was performed in triplicates.

### 2.6. RNA Extraction and Real-Time PCR

RNA was extracted from cell pellets with the RNeasy mini kit (Qiagen, Hilden, Germany) using the Qiacube automated system (Qiagen); 1 µg RNA from each sample was taken to reverse transcription reaction. Next, cDNA was generated with High-Capacity cDNA Reverse Transcription Kit (ThermoFisher Scientific, Waltham, MA, USA). Quantitative real-time polymerase chain reaction (qPCR) was performed using real time PCR (Biometra Analytik, Jena, Germany) and SYBR Green (Fast SYBR™ Green Master Mix, Applied Biosystems™, Foster city, CA, USA). Syntezza Bioscience (Jerusalem, Israel) supplied all the primers (SDF1, VEGF-A, CCL2, PDGFβ, VEGFR-2, and CXCR4). As an internal control, levels of HPRT were quantified in parallel with target genes (Appendix A, primers’ sequence). Results were analyzed using Profiler PCR Array data analysis tool (qPCRsoft3.2, Analytik Jena, Germany) and normalized to generate fold change for each gene using the ΔΔCt method [22].

### 2.7. Human EPCs Transplantation in Ectopic Subcutaneous Bone Model

In order to investigate the angiogenic potential of EPCs, cells were transplanted to an ectopic subcutaneous bone model. Beta tri calcium phosphate 0.1 g (β-TCP, Poresorb-TCP^®^, Lasak Ltd., Prague, Czech Republic) mixed with 150 μL fibrinogen (F3879, Sigma-Aldrich, St. Louis, MS, USA) and 75 μL thrombin (T7009-100, Sigma-Aldrich) served as scaffolds. Human EPCs were labeled by DII staining before transplantation in order to track their incorporation into the tissue, then incubated for 30 min in 25 µL DII stain (D282, Invitrogen, ThermoFisher Scientific, Waltham, MA, USA) and diluted in 1 mL PBS. While 2 × 10^5^ human EPCs were added to the scaffold in the test group, no cells were added to the control. Fifteen nude athymic mice (females and males, 10 weeks, 25 g) were used for transplantation, six for the control and nine for the test groups. Mice were anaesthetized with 2% Isoflurane (USP Terrell™; Piramal Critical Care, Bethlehem, PA, USA) in 100% oxygen, and scaffolds were implanted into 2–4 small 5 mm subcutaneous pouches on the back of the mice. Ten days after transplantation, fluorescein isothiocyanate-dextran (FITC Dextran; Sigma-Aldrich, Burlington, MA, USA) of 500,000 average mol wt. was dissolved in PBS to a concentration of 10 mg/mL; seconds before sacrifice, 0.2 mL were injected in the tail vein of the mice in order to label functional blood vessels in green. After sacrifice using CO2 asphyxiation, biopsies were taken and fixed in 4% paraformaldehyde for 10–20 min, then stained with NucBlue^TM^ (Molecular Probes, Eugene, OR, USA) and observed with LSM 510 Zeiss laser confocal system (Zeiss, Oberkochen, Germany).

### 2.8. Histological Preparation

Specimens were fixed with 4% paraformaldehyde before undergoing decalcification in 10% EDTA (Sigma-Aldrich) for 3 days. Specimens were embedded in paraffin, sectioned (5 μm), and stained with hematoxyline and eosine (H&E).

### 2.9. Immunohistochemistry

Slides were immuno-stained with anti-mouse CD31 (Mouse/Rat CD31/PECAM-1 Antibody, R&D systems, AF3628 Scytek, Minneapolis, UT, USA). Hematoxylin stain was used for general morphology. Blood vessels were identified as luminal perfused structures, and then quantified manually by two blinded examiners.

### 2.10. Statistical Analysis

SPSS program version 25 (IBM, Armonk, City, NY, USA) was used. Descriptive statistics included mean, standard deviation, median, and percentiles. Group comparisons were performed using a Mann-Whitney test. The linear correlation between two variables was measured by Pearson correlation coefficient. Linear regression model was used to predict blood vessel density by several independent parameters. A threshold of *p* ≤ 0.05 was set to determine significance.

## 3. Results

### 3.1. Donor Demographics

In order to compare the phenotype, genotype, and function of each individual patient’s EPCs, blood was drawn from 18 unrelated healthy donors who signed an informed consent. All donors were healthy nonsmokers, without chronic medications or history of physical trauma or surgery within the past year. EPCs were isolated and cultured as described in the methods section. Late EPCs colonies were observed from all donors. However, for only ten donors, the amount of expanded EPCs was sufficient to perform all the experiments. Therefore, this research included primary EPCs from five males and five females between 24–43 years old.

### 3.2. EPCs Expressed a High Percentage of CD31; CD34; VEGFR-2 and CXCR4

Characterization of late EPCs at passages 3–5 isolated from peripheral blood of healthy donors was performed by flow cytometry FACS analysis. Colonies of adherent proliferating cells with cobble stone morphology appeared in the culture 2–3 weeks after seeding with circulating mononuclear cells. According to flow cytometry analysis, high percentages of endothelial progenitor markers were expressed in late EPCs: CD31 (97.7 ± 3.1%), VEGFR-2 (69.5 ± 36.7%), CXCR4 (78.9 ± 35.2%), and CD34 (81.2 ± 23.5%). Low percentage of monocyte/macrophage marker CD14 (4.1 ± 4.7%) and hematopoietic marker CD45 (10.4 ± 6.5%) were observed (Table 1, Figure 1).

### 3.3. Diversity in EPCs In-Vitro Functions amongst Donors

Following characterization of EPCs from healthy donors, in-vitro assays were conducted to analyze the function of the primary isolated cells. Cell proliferation was tested using XTT assay. Equal amounts of EPCs from each donor were seeded at the start of the experiment. Optical density (OD.) was measured every 24 h for a total 72 h in quadruplicate. After calculating mean values of relative OD., the results were converted to the number of cells using a standardized calibration curve (see Appendix A). Differences in proliferation growth pattern were found among the donors. At 24 h, donors 2, 4, 7 and 10 showed an exponential increase in number of cells, whereas donors 3, 5 and 8 were at lag phase, showing no change in the number of cells. For donors 1, 6 and 9 a descended curve of cell number was found, suggesting cell death [21,23]. Cells from all donors were at log phase 48 h after seeding, indicating an elevated cell number from the previous measuring point. At 72 h, EPCs of donors 1–5 and 10 continued to proliferate, while donor 7 reached plateau. EPCs of donors 6, 8 and 9 presented declines in number of cells (Figure 2). In order to compare the proliferation rate, the donors were divided into high and low proliferative capacity according to the average number of cells at 48 h. We chose this time point specifically, as the cells from all donors were in the proliferation phase. The average number of EPCs in the high proliferation group (donors 1, 7 and 10 mean = 84266.7 ± 37975.0 cells) was significantly higher than the slow proliferating cells (donors 6, 8 and 9 mean = 17091.7 ± 6537.6 cells), **** *p* < 0.0001 (Appendix A).

In addition to cell proliferation, the chemotactic ability of EPCs conditioned medium (EPC-CM) was tested. Since successful angiogenesis requires the presence of EPCs and MSCs, we investigated the chemotactic ability of EPC-CM to enhance MSCs migration in a Boyden chamber migration assay. According to the protocol, MSCs were seeded on 8 μm porous membranes in starvation medium. EPC-CM from each donor filled the lower chamber, and the number of migrated cells counted after 12 h incubation (Figure 3A). EPC-CM increased migration of MSC compared to control by 1.5–3.5 fold. The highest chemotactic ability was found in donors 1, 3 and 5 with an average of 3.2 ± 0.3 fold and the lowest in donors 6, 8 and 9 with an average of 1.9 ± 0.4 fold. These two performances were found to be significantly different, (** *p* < 0.01) (Appendix A). The numbers of migrated MSCs toward EPC-CM from each donor were normalized relative to the number of cells that migrated towards full endothelial growth medium (EGM-2), which served as control (Figure 3B).

### 3.4. Diversity in EPCs Genotype amongst Donors

We hypothesized that high expression of specific angiogenic and chemotactic associated genes may predict angiogenesis by EPCs in-vivo. Therefore, the expression levels of SDF-1, VEGF-A, CCL2, PDGFB, VEGFR-2 and CXCR4 were analyzed and normalized to HPRT-1 (housekeeping gene) using qPCR. Relative quantification (RQ) values were normalized to donor 2, who had demonstrated average function performance in previous assays (Figure 4A). We considered high expression of a specific gene if RQ values were above 2. Average levels of highly expressed genes were demonstrated by SDF-1, PDGFB, VEGFR-2, and CXCR4. According to CXCR4 genes, highest most common expression levels were observed by donors 1, 7 and 9. On the other hand, lowest expression levels for the same gene were observed by donors 4, 6 and 10. Wilcoxon-Mann-Whitney test showed extremely significant differences for CXCR4 gene levels between the averages of two groups for high (12.5 ± 15.1) and low (0.8 ± 0.6) expression, *** *p* < 0.001 (Appendix A). These differences were not significant among the other genes: SDF-1, PDGFB and VEGFR-2, *p* ≥ 0.15. Moreover, Pearson correlation test showed significant positive correlation between the expression levels of SDF-1 and the CXCR4 (SDF-1 receptor) by all donors (R = 0.948, **** *p* < 0.0001, Figure 4B) and after excluding outlier results of expression levels for SDF-1 and CXCR4 genes (R = 0.937, ** *p* < 0.01, Figure 4C).

### 3.5. EPC Proliferation Correlates with SDF-1 and CXCR4 mRNA Levels

After analyzing all in-vitro data, we searched for associations between cell proliferation capacity, chemoattractant ability, and gene expression levels. According to Pearson correlation test, significant correlations were found between the number of cells at 48 h and mRNA levels of SDF-1 (R = 0.8001 ** *p* < 0.01) and CXCR4 (R = 0.7373 * *p* < 0.05) (Figure 5). Importantly, significant differences were found between the high to low performing donors in all in-vitro assays. Nevertheless, top performance in any single assay did not ensure superior performance in a different assay.

### 3.6. EPCs Angiogenic Capacity Varied between the Donors, However, even the Lowest Performing Donors Showed Higher Angiogenic Capacity Compared to Control

Subcutaneous mice model was used to assess the in-vivo angiogenic potential of EPCs using β-TCP scaffolds for cell delivery. All mice survived the surgical procedure without complications and were sacrificed 10 days later. Blood vessels in the scaffold stained with CD31 anti-mouse and luminal stained structures were quantified (Figure 6A). Blood vessel densities were calculated by counting the number of blood vessels in ten microscopic fields from each specimen, divided by the area of the specimen. Transplantations of β-TCP scaffolds without EPCs were referred to as control (Figure 6B). EPCs of donor 3 were not transplanted due to the limitation in cell number. Average blood vessel density in the human EPCs transplants was more than 2-fold higher compared to control (134.1 ± 16.8 vs. 79.4 ± 14.0 **** *p* < 0.0001). Donors 4, 5 and 7 show the highest blood vessel densities, with an average of 163.3 ± 17.2. The proangiogenic capacity of EPCs was also found in the donors with the lowest average blood vessel density (donors 2, 6 and 8). The average blood vessel density in the latter group (108.4 ± 19.1) was still significantly different from control (79.4 ± 14.0, * *p* < 0.05). Comparing blood vessel density between the donors with the high and low performances showed extremely significant difference between these groups, **** *p* = 0.0001 (Figure 6C).

EPCs play dual effect in angiogenesis: the main mechanism is paracrine effect by recruiting resident mouse endothelial cells, as seen in Figure 6A, using anti-mouse CD31. The second mechanism is incorporation of EPCs into blood vessel walls. In order to track the cells in the tissue and follow their direct engraftment in blood vessels walls, cells were labeled with fluorescent dye before transplantation and vessels were perfused with FITC dextran. Figure 7 demonstrates the engraftment and incorporation of EPCs in blood vessels walls (Figure 7A,B).

### 3.7. Prediction of In-Vivo Blood Vessel Formation Using a Regression Model Equation

Heterogeneity in EPCs angiogenic function between donors may influence the expected results following cell transplantation, and preclude standardized use of EPC therapy. Ideally, finding in-vitro tests that can predict the in-vivo angiogenic capacity of EPCs obtained from unrelated donors would be a significant breakthrough in this field. Using Pearson correlation test, we analyzed the association between in-vivo blood vessel density and in-vitro assays: proliferation rate, chemo-attractive ability, and levels of gene expression.

We failed to find a significant association between these parameters when analyzing the whole group. However, analysis of the results of donors within the 50th percentile enabled the performance of a linear regression model that generated a mathematical equation to predict blood vessel density. The equation relies on EPCs chemoattractant ability and VEGFR-2 mRNA levels as follows, where A represents the blood vessel density (dependent variable); B represents the fold changes in MSCs migration to EPCs condition medium (independent variable), * *p* = 0.036; and C represents the expression level of VEGFR-2 gene (independent variable), * *p* = 0.032. According to the equation, it can be noted that variable C is more dominant than variable B, since it has a higher standardized coefficient Beta (β2 = 0.814 vs. β1 = 0.77) and higher relevant factor (78.47 vs. 58.89) (see Table 2):A = −17.1 + 58.89∙B + 78.46∙C(1)

According to the equation, it can be noted that variable C is more dominant than variable B, since it has a higher standardized coefficient β (β2 = 0.814 vs. β1 = 0.77) and higher relevant factor (78.47 vs. 58.89).

Donors #7 and #10 had similar migration (B) but differed in VEGF-2 expressions (higher in donor #7). Therefore, blood vessel density (A) was higher in donor #7, who also presented with higher blood vessel count in–vivo.

Donor #5 had higher chemoatractive ability (B) and higher VEGFR-2 (C) compared to donor #10. The calculated blood vessels (A) and counted blood vessels in vivo were higher for donor #5 compared with donor #10.

## 4. Discussion

A hallmark of progenitor cells is their ability to proliferate and differentiate into mature cells; as such EPCs have high replication capacity and differentiate to mature endothelial cells. EPCs efficiency have been demonstrated in cell-based therapies, when improved vasculogenesis/angiogenesis in multiple therapeutic applications [24,25] and local transplantation of autologous EPCs into bone defects resulted in significant increase in bone formation [10,26,27].

One of the impediments using autologous cell transplantation is variation in cell function that arises from genetic, systemic, and environmental differences between patients. These variations may affect the final outcome following cell therapy, giving less than optimal results in some patients. In this study, variances in EPCs performances were observed among the donors from both in-vitro and in-vivo assays. We utilized a linear regression model and were able to predict the in-vivo angiogenic potential of EPCs isolated from unrelated healthy donors based on in-vitro chemo-attractive ability and VEGFR-2 expression levels.

An additional obstacle in EPCs research is the absence of standard protocol for cell characterization. EPCs represent a heterogeneous population of cells lacking precise phenotype. Due to EPCs heterogeneity and the disparate results in different clinical studies, the true identity of these cells remains elusive. For example, Kaur et al. [28] isolated EPCs from the peripheral blood of patients with liver cirrhosis. The researchers used isolation and culture protocols similar to ours. Cultured EPCs were identified by flow cytometry for CD45, CD34, CD31, and KDR followed by detailed phenotyping using CyToF. Relying on both methods, the authors identified three distinct populations of circulating EPCs classified by their CD45 expression (negative, intermediate, and high). In another study, Dauwe et al. [29] isolated and cultured EPCs from patients with stable ischemic cardiomyopathy. Isolation and culture protocols were similar to ours. Flow cytometry results showed high expression levels of CD31 with variability in CD34 and VEGFR-2 among the participants. Attempting to optimize FACS protocol for EPCs characterization, Huizer et al. [30] isolated EPCs from 18 blood samples (using identical isolation protocol). Generation of late EPCs (proliferating outgrowth) were detected in only two samples. The cells displayed cobblestone morphology and expressed high and stable levels of CD31with heterogeneous expression of KDR and CD34. The large variability in the expression of CD45, KDR, CD34 might originate from genetic and or environmental differences among humans, or be due to differentiation of the cells in culture during passaging or the presence of early and late EPCs in the sample. In the current study, we used cells in passage 3–7; the expression levels of any of the examined membrane markers were not associated with regard to in-vivo angiogenesis. Currently, late outgrowth EPCs are commonly identified by their phenotypic distinct properties for producing colonies in culture, in addition to co-expression of endothelial and hematopoietic markers. They were also described as homogeneous endothelial-like progenitor cell population that possess a high proliferative potential, differentiating into vascular endothelial cells, and forming networks in-vitro and in-vivo [31,32,33].

Cell proliferation capacity is one of the most common functional assays that was found to correlate with in-vivo function [34]. According to Janicki et al., a doubling time below 43 h allowed to predict ectopic bone formation by MSC at high sensitivity and specificity. In this study, at the first 24 h, for most of the donors, cells were in the lag phase probably due to recovery after trypsinization [35]. Between 24 to 48 h, cells from all the donors were in the log phase, with an average of 1.5 fold-change during this period. Moreover, the mean doubling time at the log phase for all the donors was 39.4h. According to Deskins DL et al. [36], when MSCs were isolated from bone marrow samples of 10 healthy individuals undergoing orthopedic surgeries, these cells presented an average of 1.3 folds change in cell growth. Likewise, cell proliferation study performed during the logarithmic phase of human umbilical vein endothelial cell (HUVEC) growth demonstrated a doubling time of 92 h [37]. Comparing our findings to these results, it is suggested that human EPCs growth rate is higher than that of MSCs and HUVECs. Doubling time of human embryonic stem cells (hESCs) is shorter than for human EPCs (28 h vs. 39.4 h, respectively) [38]. This difference can be expected, since hESCs are the most primitive stem cell population that can self-renew indefinitely when maintained in appropriate conditions

The potential of EPCs to promote angiogenesis is enhanced by the presence of MSCs. Therefore we hypothesized the chemotactic ability of hEPC to recruit MSCs might be a predictor of the angiogenic potential of EPCs. In-vitro, co-culture of EPCs with MSCs demonstrated enhanced cell proliferation and angiogenesis [39,40]. Researchers investigated the paracrine effect of hEPCs and mouse MSCs-CM on endothelial cells and found a combined CM promoted endothelial cells adhesion and proliferation in vitro [41]. According to our results, EPCs-CM promoted the migration of MSCs with varied intensities between donors. Likewise, the chemoattractive ability of MSC-CM was influenced by donor variability and tissue origin (bone marrow vs. adipose tissue) [42]. This heterogeneity may affect the results of EPCs or MSCs transplantation in animal models and clinical trials.

Another factor associated with donor variability that might affect cell function is cell genomic profile. The immune potency of MSCs was investigated in patients with Chron’s and graft versus host disease. A quantitative RNA-based array for genes specific to immunomodulatory and homing properties of MSCs was generated to predict the immunogenic potential of the cells [43]. As such, we intended to evaluate the genomic profile of EPCs from healthy subjects, focusing on proangiogenic and chemoattractant genes. Using Real-Time PCR, we found differences in RQ values distribution for all the analyzed genes. An interesting outcome was found among SDF-1 chemokine and its receptor, CXCR4, for having a positive correlation between their expression levels (R = 0.948, *p* < 0.0001). These genes were also in a positive correlation with EPC proliferation. This result can be clarified by a previous study, which suggests that SDF-1 may be secreted by hematopoietic stem/progenitor cells and involved in autocrine/paracrine regulation of their development and survival [44]. Other studied have also shown that adeno virus gene transfer of SDF-1 enhances the number of circulating hematopoietic stem cells/EPCs [45,46,47].

An additional report presenting SDF-1 regulation of EPCs proliferation and migration through CXCR4 and CXCR7 receptors in-vitro, showed a positive dose–response effect of SDF-1 on EPCs proliferation. Furthermore, EPCs pretreated with SDF-1 showed elevated cell proliferation and migration that was suppressed following CXCR7 and CXCR4 blocking [47].

Interestingly, neither gender nor the age of the donor influenced EPCs in-vitro function or genes expression levels (data in Appendix A). On the contrary, a study of 20 young (25 ± 1 year) vs. 20 old (61 ± 2 years) healthy subjects, investigated alterations in EPCs functions by age. Study results showed lower survival, migration, and proliferation potentials in EPCs from old subjects. These findings indicate that maintenance of vascular homeostasis by EPCs may be attenuated with maturity, based on functional deficits [48]. Another study of patients (43–80 yrs.) with stable coronary artery disease undergoing coronary artery bypass grafting revealed that age is a major limiting factor for EPCs mobilization [49]. The inconsistency in the results between these studies to ours can be attributed to the relatively young age (24–43 years) of the participants in the present research, compared to wider age distributions in some other studies.

In order to evaluate EPCs pro-angiogenic function, we evaluated blood vessel formation in-vivo. EPCs from healthy donors were seeded on β-TCP scaffolds and transplanted into ectopic subcutaneous mice model. Transplantation of β-TCP scaffolds alone served as control. A previous study showed superior adhesion and proliferation of human bone marrow MNCs to β-TCP compared to other scaffolds [50]. Subcutaneous implantation is one of the most common technique in animal models to evaluate angiogenesis [51,52]. The results of this study showed differences in blood vessel density between the donors. Nevertheless, even donors demonstrating the lowest blood vessels densities were still significantly higher than control group. Therefore, clinically we can expect different outcomes following autologous EPCs transplantation even among young healthy individuals (such as patients suffering from bone trauma). Moreover, our findings emphasize the need for discovering in-vitro potency assay that can predict EPCs in-vivo function. The potential aspect of autologous cell therapy in patients and its efficacy is still controversial. Previous reports concerning treatment of pediatric diseases based on MSCs emphasize the diversity of this host response effect. A study of osteogenesis imperfect in pediatrics patients who received allogeneic bone marrow transplants from HLA-matched siblings showed significant improvement. Nevertheless, since this treatment is not sustainable, patients underwent intravenous treatment with cultured bone marrow MSCs after the original transplantation. While five patients showed improvement in growth velocity, one patient experienced toxicity related to MSC infusion [53]. A more recent study of children with acute grade III–IV graft-versus-host disease (GvHD) treated with allogeneic bone marrow MSCs, showed 65% complete response and significantly better overall survival at 6 years, while 22% had a partial response [54]. Based on this approach, we can conclude that cell therapy in clinical research is unpredictable, probably due to differences in the function of transplanted cells between donors.

To investigate a relationship between EPCs in-vitro and in-vivo function, we initially used Pearson correlation test for blood vessel density, corresponded with cell proliferation, chemo-attractive capacity, and gene expression. None of the tested parameters were associated when the whole group was included. Utilizing a linear regression analysis only for the donors whose performances were in the 50 percentile (excluding the top and lowest donors) we were able to predict blood vessel density by two independent in-vitro variables: (i) fold changes in MSCs migration to EPCs condition medium, and (ii) expression level of VEGFR-2 gene. While the latter parameter was more dominant, VEGFR-2 is considered a major positive signal transducer for both physiological and pathological angiogenesis. According to research, knock-out mice for VEGFR-2 gene led to their death due to lack of vasculogenesis [55]. Studies declared that VEGF stimulates endothelial cell migration and angiogenesis mainly through VEGFR-2 [56,57]. This indicates that VEGFR-2 signaling is essential for the proliferation and differentiation of VEGFR-2-positive endothelial precursor cells into vascular endothelial cells [13]. The correlation between the number of formed blood vessels and chemo-attractive effect of EPCs condition medium on MSCs is in agreement with previous studies, which claim that EPCs strong paracrine effect is the main mechanism for angiogenesis and wound healing [58,59].

Regardless of the novelty of the results, there are several limitations to this study. First, we were able to isolate and culture EPCs from peripheral blood in only 10 out of 18 donors. The use of G-CSF might have increased the chances to isolate this rare population of cells from the blood, but not without significant risks. Therefore, in clinical trials (but not for research purposes) this strategy should be considered. In the future, we will try to add SDF-1 to the culture medium, as it was in correlation with EPCs proliferation. An additional drawback of the study is the small and relatively homogenous research population. However, other studies aimed at predicting the potency of human MSCs used 6-10 donors. Worth mentioning, MSC isolation and expansion is much more predictable compared to EPCs expansion. Deskins et al. [36] isolated MSC from the bone marrow of 10 human donors and performed in-vitro assays to evaluate cell proliferation and viability as well as in vivo cell transplantation to evaluate the engraftment of the cells. Samsonraj et al. [60] established criteria for human MSC potency using bone marrow biopsies from six healthy donors. Recently, Wang et al. [61] compared the potency of MSCs isolated from six patients suffering from chronic pancreatitis versus cells from two healthy donors. In the future, we intend to expand the study to a larger cohort with a wider age range and different systemic status in order to shed light on the therapeutic potential of EPCs for these patients, and bring us incrementally closer to personalized medicine.

## 5. Conclusions

EPCs isolated from unrelated healthy donors demonstrated heterogenic in-vitro and in-vivo functions. It is critical to declare that despite several donors who demonstrated superior results in all tests, top performance of an individual in any single in-vitro assay, did not necessarily predict superior performance in other assay. Prediction of blood vessel formation by EPCs is essential for clinicians using autologous cell therapy for the treatment of ischemic conditions, wound healing, and bone repair. In the future, it may be possible to utilize in-vitro tests, such as VEGFR-2 expression and chemo-attractive capacity, to predict angiogenesis following EPCs transplantation. Developing a potency kit may be a useful tool in personalized medicine but will require validation of results in a larger sample group Despite the variability among the donors, our findings illustrate a positive impact of human EPCs on angiogenesis.

## Figures and Tables

**Figure 1 biomolecules-09-00717-f001:**
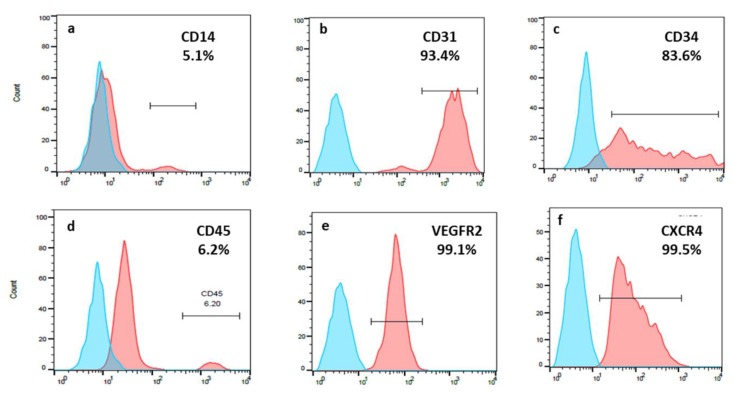
Flow cytometry analysis of primary EPCs: gated population of EPCs from donor 2, typical fluorescence in forward and side scatter, histogram representation includes unstained sample as a reference, revealing the percentage of positive stained cells.

**Figure 2 biomolecules-09-00717-f002:**
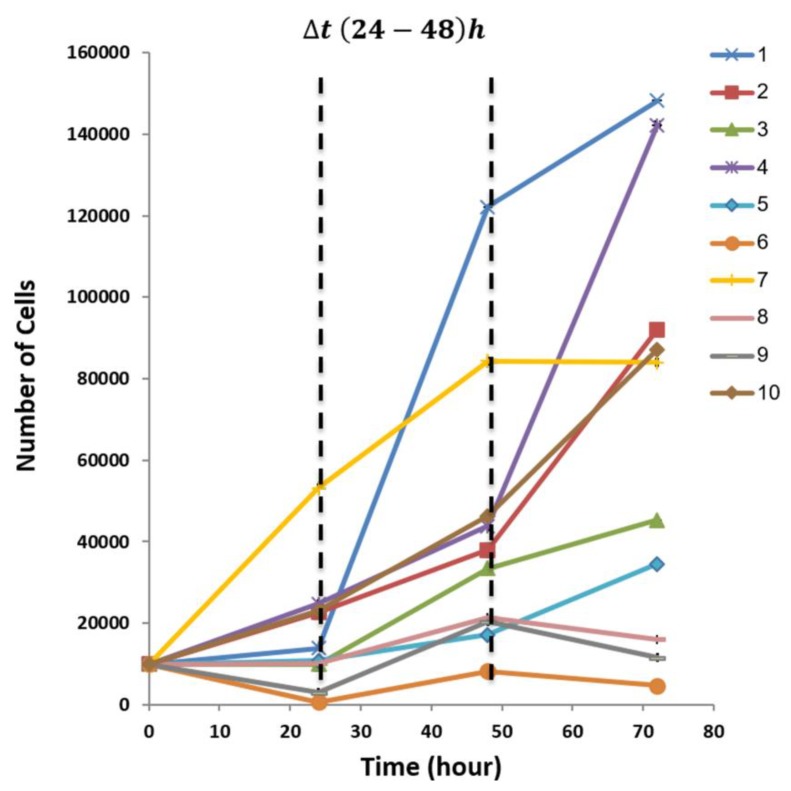
EPCs proliferation rate. Evaluation of the cell count was performed at four time points using XTT assay.

**Figure 3 biomolecules-09-00717-f003:**
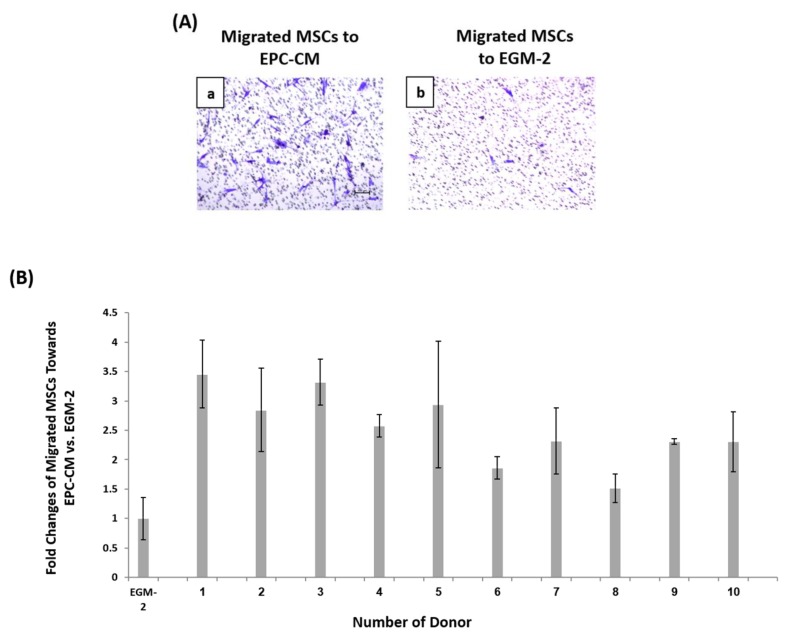
EPCs chemoattractant ability. (**A.a**) Migrated MSCs towards EPCs condition medium and (**A.b**) growth media EGM-2. (**B**) Fold changes of migrated MSCs to donors’ EPCs condition medium relative to EGM-2.

**Figure 4 biomolecules-09-00717-f004:**
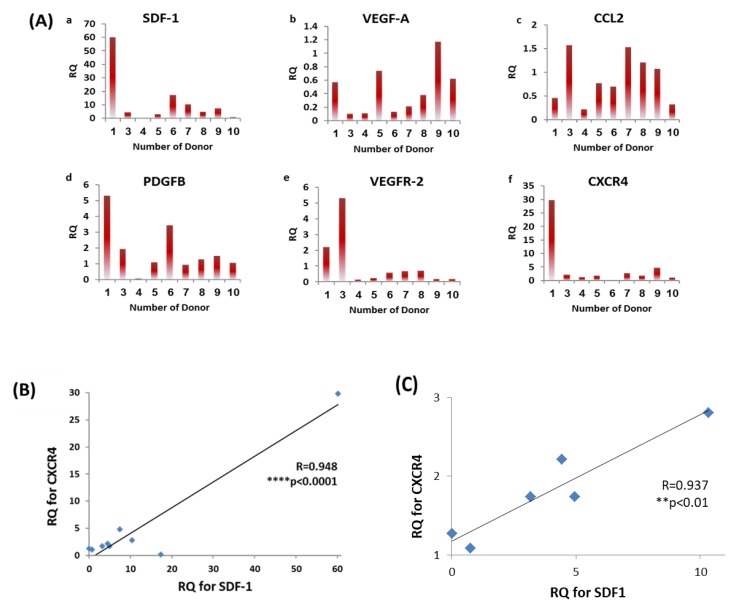
EPCs expression levels of angiogenic and chemoattractant genes. (**A**) mRNA expression of (**a**) SDF-1 (**b**) VEGF-A (**c**) CCL2 (**d**) PDGFB (**e**) VEGFR-2 (**f**) CXCR4 genes. RQ values were normalized to donor 2. (**B**) Positive correlation between SDF-1 and CXCR4 expression levels between all donors, R = 0.948 **** *p* < 0.0001. (**C**) and after excluding outlier results, R = 0.937 ** *p* < 0.01.

**Figure 5 biomolecules-09-00717-f005:**
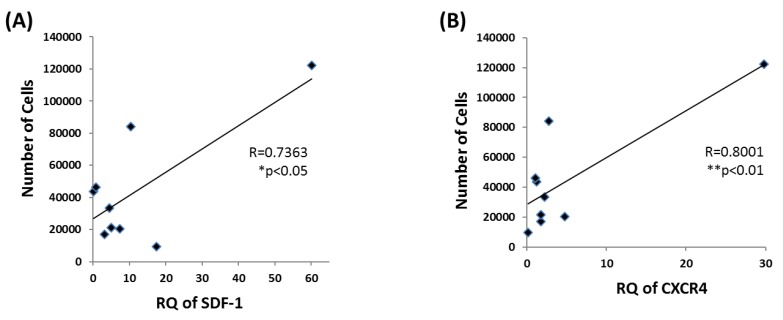
Correlation between EPCs proliferation rate and SDF-1 and CXCR4 genes. Cell number after 48h demonstrated a positive correlation with expression levels of (**A**) SDF-1 gene, R = 0.8001 ** *p* < 0.01, and (**B**) CXCR4 gene, R = 0.7373 * *p* < 0.05.

**Figure 6 biomolecules-09-00717-f006:**
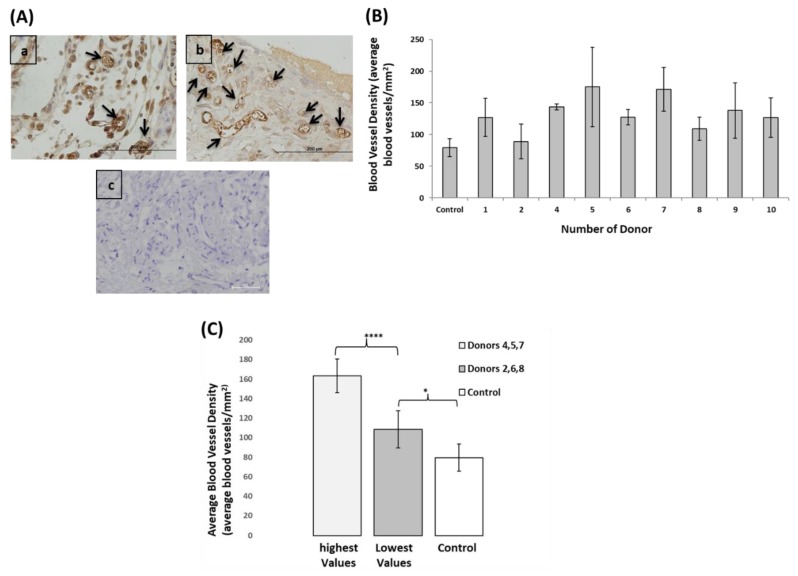
In-vivo angiogenic potential of EPCs (**A**) Histological analysis of EPCs subcutaneous transplantations. CD31 anti-mouse staining of in-vivo subcutaneous transplantation β-TCP scaffold (**a**) without cells for control (**b**) with human EPCs (**c**) secondary antibody control. Functional blood vessels are indicated by a black arrow. (**B**) Blood vessel density in the subcutaneous implants. (**C**) Comparison between donors with high vs. low blood vessel densities relative to control **** *p* < 0.0001 and * *p* < 0.05, respectively.

**Figure 7 biomolecules-09-00717-f007:**
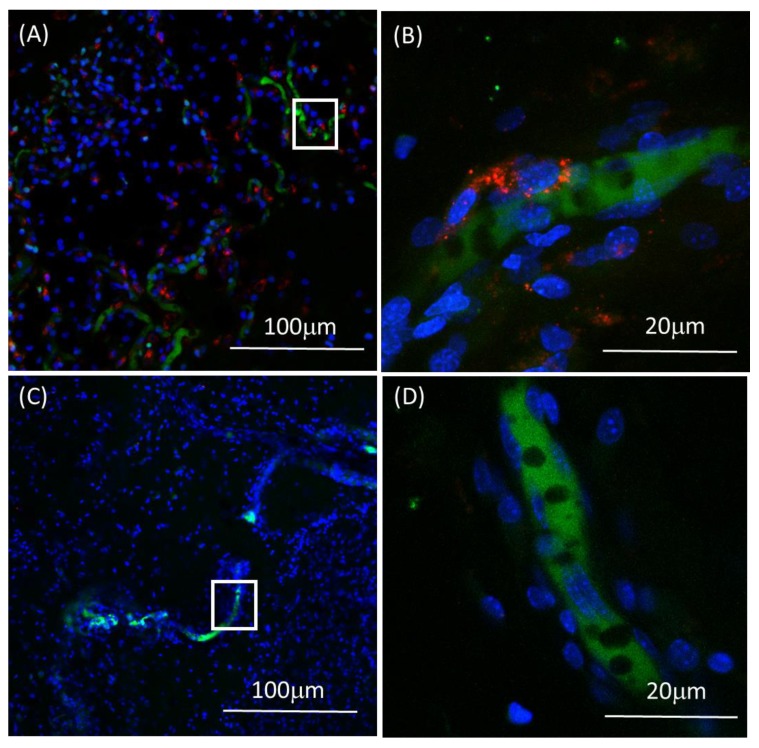
Tracking transplanted EPCs 10 days after transplantation in a mice model. (**A**) Test sample, EPCs (red) were detected in the new tissue. (**B**) Higher magnification of A showing EPCs (red) lining blood vessel lumen (green). (**C**) Control sample, blood vessels stained in green and the endogenous cells stained blue. (**D**) Higher magnification of C.

**Table 1 biomolecules-09-00717-t001:** Flow cytometry analysis of primary EPCs. Quantitative FACS analysis of EPCs isolated from all donors.

Number of Donor	Positive Stained EPCs (%)					
	CD14	CD31	CD34	CD45	VEGFR-2/KDR	CXCR4
1	1.9	98.8	98.9	18.1	69.2	99.7
2	5.1	93.4	83.6	6.2	99.1	99.5
3	0	99.2	93.0	3.8	96.5	98.3
4	1.8	100	55.4	1.9	98.7	98.0
5	4.8	99.6	84.7	14.3	22.8	97.3
6	1.9	93.8	96.2	5.9	8.9	2.8
7	0.6	100	21.5	3.1	63.3	67.3
8	5.7	91.7	76.9	10.5	15.8	18.4
9	17.0	99.4	98.8	20.6	99.0	90.4
10	3.7	99.7	94.1	14.4	100	96.2
Average ± SD	4.1 ± 4.7	97.7 ± 3.1	81.2 ± 23.5	10.4 ± 6.5	69.5 ± 36.7	78.9 ± 35.2

**Table 2 biomolecules-09-00717-t002:** Regression analysis calculations for representative donors and actual blood vessels counts in the mice model for the same donors.

Donor	Migration (B in Equation)	RT-PCR VEGFR-2 (C in Equation)	Calculated Blood Vessels (A in Equation)	In-Vivo Vessel Count
5	2.94	0.23	173.96	175.0
7	2.32	0.66	171.10	171.2
10	2.30	0.17	131.98	126.8

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
