# Peer review of "Predicting Angiogenesis by Endothelial Progenitor Cells Relying on In-Vitro Function Assays and VEGFR-2 Expression Levels"

_biomolecules, 2019, doi:10.3390/biom9110717_

Round 1

Reviewer 1 Report

Sabbha et al describe the prediction of blood vessel formation in autologous endothelial progenitor cell (EPC) transplant. Autologous transplantation of EPCs is a new strategy to achieve therapeutic angiogenesis in humans. The study evaluates how genetic differences influence therapeutic outcome and concludes that in vitro tests such as VEGFR 2 expression and chemo-attractive capacity can be used to predict angiogenesis following EPC transplantation.

Comments:

The manuscript needs to be edited by native English speaker.

Typo-page 3, 2.2: 2X105 should be 2X105

Page 3, 2.6: references needed for Profiler PCR Array data analysis tool and △△Ct method.

Reviewer 2 Report

Angiogenic activity of EPCs is important for the recovery of damaged organs, such as infracted myocardium, and has potential therapeutic effects on several ischemic diseases. However, the level of the activity varies among individual EPCs. Thus, in this study, the authors tried to precisely predict the angiogenic activity of EPCs obtained from healthy donors. The in vivo blood vessel formation after the subcutaneous transplantation of EPCs into the mice was likely to rely on i) EPCs chemoattractivity in in vitro experiments and ii) VEGFR2 mRNA levels. In general, however, there are several critical issues that remain to be elucidated to draw the conclusion, and the results shown in the manuscript are too preliminary and immature.

1. Based on the results of FACS in Fig. 1A, EPCs derived from 10 healthy donors appeared too heterogeneous. For example, EPCs from #7 donor had low expression of CD34, and EPCs from #9 donor showed high expression of CD14 and CD45 (including hematopoietic cells?), compared with those from other donors. Thus, the scientific significance of comparison of such EPCs might be questionable. In addition, was the experimental technique to expand EPCs appropriate?

2. The comparison between highest and lowest values in Fig. 2B, 3C, 4B and 6C has little significance.

3. The correlation in the graphs of Fig. 4C and 4 seems overestimated, because only one outlier in each graph induced the statistical significance for the correlation. If the outlier is excluded from each graph, it would be quite difficult to observe the correlation.

4. In the EPCs subcutaneous transplantation assay, it is uncertain whether the neovascularization in the beta-TCP scaffold is formed by EPCs themselves or is derived from the mouse blood vessels around the scaffold.

5. Although the authors insist that the VEGFR2 mRNA level essentially contributes to the prediction of angiogenesis, EPCs from #1 and #3 donors, which expressed the higher amount of VEGFR2 mRNA than those from others in Fig. 4A, did not induce the high blood vessel density in Fig. 6 (EPCs from #3 were not used in the in vivo experiment due to a few numbers of cells). From these results, it is very hard to understand the importance of VEGFR2 mRNA level for the EPCs angiogenic activity.

Reviewer 3 Report

Overall the idea and aim proposed is good but the article needs improvements.

The major drawback is the sample size. Ten samples seems to me quite a low number to infer good conclusions. Besides, the authors could only work with 10 out of 18 donors, which is only 55% efficiency of EPC’s isolation from patient’s blood.

Another point that needs improvement is the age of the donors. Conclusions were taken in an age group between 24-43 years old. In practice, would EPCs always be isolated from “younger” donors?... Sample size should be increased, as well as age range of the donors. All negative associations found might have to do with the age range of the group studied.

There are some spelling and english mistakes throughout the paper:

In page 2, there are “;” instead of “.” (first line); there is “felids” instead of “fields” (line 3). Second phrase of page 2 needs to be clear: “as long-term safety”?? Same phrase should have either “recently described” or “were published”.

Second phrase from second paragraph on page 2 is confusing…. First phrase needs reference. In the last phrase of second paragraph should be “was” instead of “were”.  

Page 2, second paragraph, first phrase lacks references.

Throughout the text there is the need of correcting the writing: there are unnecessary commas, few spelling mistakes, lack of spaces, when mentioning cell number the exponent should always be superscript, etc…

Page 3, point 2.2, rephrase “percentage calculation”. Point 2.3, why is “O.D.” in bold? And “Talble 1S” should be between parenthesis. Point 2.5, should be “a” 8mm instead of “the”.

Page 4, first paragraph: where is Table 2??

Point 2.7: is it Mice Model?? Or Bone Model??

Page 6, first paragraph: refer the figure number in the supplementary file.

The authors say there was a “slight difference” in the proliferation growth pattern between the donors. Was it slight?

Figure 3: figure B should come before figure C

Page 7: expression leelvs are normalized instead of calibrated to.

Discussion is a bit confusing at some paragraphs.

This study presents several limitations and thus the conclusion is a overstated.

Round 2

Reviewer 2 Report

After reading the re-submitted manuscript, this reviewer considers that the improvement is too little, compared with the original manuscript, and that several critical issues still remain unanswered, although some data have been displayed additionally.

The main aim of this study is to originally find the general role to predict the ability of EPCs in angiogenesis for the advance of EPC therapy. For this aim, using heterogenic EPCs has less scientific meaning. At least, amplified EPCs have to be sorted as the fraction positive for CD31 and CD34 and negative for CD45, and then, the sorted cells are utilized for further experiments.

The authors did not appropriately respond to the major comments 2 and 3.

Author Response

4.11.19

Response to Reviewer 2

Comment 1:

The comparison between highest and lowest values in Fig. 2B, 3C, 4B and 6C has little significance.

Response 1:

Figures 2B, 3C and 4B were deleted from the manuscript and transferred to the supplement file (Fig. 2SA-C). Figure 6C demonstrates a comparison between test (TCP+EPC) and control (TCP) therefore was not omitted.

Comment 2:

The correlations in the graphs in Fig. 4C and 4 seem overestimated, because only one outlier in each graph induced the statistical significance for the correlation. If the outlier is excluded from each graph, it would be quite difficult to observe the correlation.

Response 2:

We repeated the correlation between SDF-1 and CXCR4, excluding outliers. The correlation was positive and significant: R=0.9367, P=0.0058. The graph without outliers was added to the manuscript (Fig 4C).

Reviewer 3 Report

The authors have agreed with the given suggestions and the paper looks better.

There are, however, still some minor reviews that should be made, mainly concerning editing.

Line 38 – reference missing at the end of the phrase. Although there are references further ahead in the text, there should be a reference here.

Line 42, 107, 111, 137, 236, 373, 411, 413, 418, 432, 436, 440, 465, 513, 830 – edit spaces or lack of them

Line 53 – no comma after “status”

Line 111 – “…relative to according to the slope…” ?? Rephrase please.

Lines 139 and 141 – in indeed BONE model??

Line 191 – FACS should be before “analysis”

Line 198-199 – Should not be Figure 1 A and B, but Table 1 (since it is a table, not a figure) and figure 1. Change in the text accordingly.

Line 239 – The p value should be between ( ).

Line 266 – p and R values should be between ( ).

Line 315 – The same as I have told previously: figures should be presented sequentially – change B with C and in the legend/text accordingly

Line 357 – Would you accept my suggestion in lane 198-199, this should now be table 2

Lines 384 to 405 – Haven’t checked references….

Line 498 – “…common TECHNIQUE IN animal models…”

Didn’t go through all references. Please check them.

Author Response

Response:

Line 38- References were added.

Line 315 – The same as I have told previously: figures should be presented sequentially – change B with C and in the legend/text accordingly-Amended

Typos were corrected and spaces deleted.

Figure 1 was separated to: Fig. 1 and Table 1

References were checked and corrected.